# Homogenization of Functional Diversity of Rotifer Communities in Relation to Eutrophication in an Urban River of North China

**DOI:** 10.3390/biology12121488

**Published:** 2023-12-04

**Authors:** Bing Wang, Xuwang Yin

**Affiliations:** Liaoning Provincial Key Laboratory for Hydrobiology, College of Fisheries and Life Science, Dalian Ocean University, Dalian 116023, China; wang18047298715@163.com

**Keywords:** biotic homogenization, rotifer, functional group, functional diversity, beta diversity

## Abstract

**Simple Summary:**

Eutrophication is a significant challenge for urban aquatic ecosystems. Many studies suggest the negative effect of eutrophication on the diversity of planktonic communities; however, we still know little about the characteristics of aquatic functional groups in relation to varying degrees of eutrophication in urban river systems. The effects of urban river eutrophication on rotifer communities are investigated using an annual field survey of the Jinan section of the Xiaoqing River, a typical urban river in northern China. We observed that the functional diversity of the rotifer community decreased with the degree of eutrophication. Functional diversity exhibited an extremely low level across functional groups. These findings indicate that eutrophication led to the homogenization of rotifer communities, which can be attributed to the widely distributed species complementing the ecological niche space. Our findings provide valuable information on the conservation of the urban river under the threat of eutrophication caused by high-intensity human activities.

**Abstract:**

Rapid urbanization has triggered nutrient loading, which will inevitably lead to the eutrophication of water bodies and further affect the structure of aquatic populations. At present, eutrophication is a significant challenge for urban aquatic ecosystems. However, we still know little about the correlation between eutrophication in urban rivers and the composition of aquatic functional groups. The effects of urban river eutrophication on rotifer communities were investigated using an annual field survey of the Jinan section of the Xiaoqing River, a typical urban river in northern China. Using functional diversity (FD) and beta diversity, the spatiotemporal variation of the aquatic biological functional groups regime along stretches subject to different eutrophication was investigated. The functional evenness (FEve) and functional divergence (FDiv) decreased significantly with the increment of the trophic level index. Functional diversity exhibits an extremely low level across functional groups, with the richness difference (RichDiff) being an important component. The results indicate that eutrophication led to the homogenization of rotifer communities. This can be attributed to the functional homogenization of the rotifer community in the Jinan section of the Xiaoqing River. The observed homogenization may be due to widely distributed species complementing the ecological niche space. Our findings provide valuable information on the conservation of the urban river under the threat of eutrophication caused by high-intensity human activities.

## 1. Introduction

High-intensity human actions have become a major driver of global environmental change [1]. As an important natural resource for sustainable human development, aquatic ecosystems are under tremendous pressure [2]. Urbanization has been significantly accelerated by the significant increase in the proportion of the urban population worldwide [3,4]. Rapid urbanization may affect the chemical composition of aquatic ecosystems, especially the loading of nitrogen and phosphorus, which can lead to an imbalance in their natural dynamics, and loads exceeding the assimilative capacity of aquatic ecosystems can lead to eutrophication [4,5]. Eutrophication further affects the structure of aquatic populations, leading to biodiversity loss and ultimately threatening the quality and stability of aquatic ecosystems [6,7].

As a link to aquatic ecosystems, rivers play a key role in the material cycle and energy flow channels [8,9]. Urban rivers are characterized by a high degree of artificiality and are more susceptible to anthropogenic pressure [10,11]. Other similar features, such as eutrophication and biodiversity loss, are present in urban rivers worldwide and are referred to as “urban river syndrome” [2,7,11,12]. Numerous studies have shown that eutrophication in urban rivers causes the loss of aquatic organisms’ biodiversity and that communities tend to be homogenous at the species level. However, little has been performed to assess the effects of eutrophication on the composition of aquatic functional groups in urban rivers [13,14,15].

Homogenization is a process that reduces biological differences among biotas at specific time intervals [16,17]. Eutrophication caused by human activities is considered one of the most important causes of biotic homogenization [18]. Eutrophication reduces habitat and lowers biodiversity, which provokes intensified competition for a few shared niches [19]. Biotic homogenization due to an increased similarity in community composition may significantly affect ecosystem stability [18]. Furthermore, homogenization due to eutrophication occurs at the taxonomic level and involves functional homogenization as species adapt to environmental changes concerning their functional traits [20]. Functional homogenization, as a subtype of biotic homogenization, is a process wherein the functional trait composition of communities tends to become more similar [21]. Functional diversity serves as a linkage between species traits and ecosystem dynamics by facilitating complementary resource use. It stands as an essential feature within biological assemblages [22,23]. Therefore, investigations focusing on the response of functional group structures within communities to environmental changes can offer insights into the significance of maintaining biodiversity for ecosystem stability [24].

Eutrophication functions as an environmental filter, favoring species with similar traits that can adapt to environmental changes associated with urbanization, thereby contributing to biotic homogenization. Simultaneously, it influences beta diversity in riverine ecosystems [25,26]. Functional beta diversity plays a crucial role in elucidating the intricate process of community assembly under the influence of filtering environmental factors [27]. Total beta diversity consists of components of two distinct ecological processes: RDiff (species richness difference and the loss or gain of species between sites) and species replacement (replacement and species replacement between sites) [28,29]. Low differences in functional abundance suggest a convergence of species from each community in terms of functional and niche assimilation between communities [30]. The response of functional groups to homogenization may give rise to a pronounced substitution pattern, leading to reduced functional beta diversity within the community. Simultaneously, a high similarity between communities may compromise ecosystem functional diversity [31,32]. Hence, β-diversity can serve as a tool to explore the impact of eutrophication on riverine ecosystems, assessing its potential role in inducing functional homogenization [26].

Rotifers were selected as model organisms in the present study to examine the structural ramifications of eutrophication in urban rivers on aquatic biomes. Rotifers are vital zooplankton in aquatic ecosystems [10]. They are commonly recognized as effective indicators for evaluating the trophic levels of aquatic organisms due to their sensitivity to alterations in the nutritional status of the environment. Furthermore, the composition and structure of the rotifer’s community exhibit significant fluctuations in response to environmental changes [33,34,35]. In the current study, the temporal dynamics of environmental factors and rotifer communities in an urban river at a regional scale were investigated, and the following hypotheses were tested: (1) eutrophication would gradually homogenize rotifer communities and (2) the homogenization in a community might be functional homogenization and the result of widely distributed species complementation.

## 2. Materials and Methods

### 2.1. Study Sites

The Xiaoqing River originates in Jinan City and flows in a west-to-east direction, ultimately discharging into Laizhou Bay of the Bohai Sea. Serving as the sole drainage canal in the main urban area of Jinan, human activities concentrated along its course have significantly impacted the local ecosystem. This study was conducted at ten sampling sites within the Jinan section of the Xiaoqing River, with monthly sampling carried out at each site from April 2020 to March 2021 (Figure 1).

### 2.2. Sampling and Analytical Procedure

The latitude, longitude, and altitude of the sampling sites in the Jinan section of the Xiaoqing River were recorded using the MAGELLAN global positioning system (eXploist-200, Magellan Aerospace, Glendale, CA, USA). On each sampling date, 50 L of water was collected at a depth of 1.5 m at each sampling site using a 5 L specialized sampler. The collected samples were promptly preserved with a 1.0–1.5% Lugol solution, and the supernatant was concentrated to 30 mL after settling in a cylindrical separatory funnel for 24 h to quantify rotifers. Following mixing, 1 mL of the concentrated sample was randomly extracted for full-slide observation under an Olympus BH-2 microscope (Olympus, Tokyo, Japan) at 100× magnification, and this process was repeated twice. Species identification was conducted based on the guidelines outlined in Chinese freshwater rotifer references and Guides to the Identification of the Microinvertebrates of the Continental Waters of the World.

Physical factors, such as water temperature (Temp), water depth (Dep), water transparency (SD), dissolved oxygen (DO), and pH, were measured in situ at the sampling sites. Water samples (2 L) were collected at each sampling site and stored frozen. Chlorophyll-a (Chl-a), chemical oxygen demand (COD), total dissolved phosphorus (TP), ammonium nitrogen (NH4-N), and total dissolved nitrogen (TN) were determined in the laboratory using the frozen water samples using standard analytical methods [36].

### 2.3. Functional Group Categorization

To analyze correlations between the rotifer community composition and eutrophication in urban rivers, a functional classification was created based on species characteristics, which were classified into functional groups based on their trophic level, size, feeding habits, and swimming (Table 1). According to the “Rotifer trophi web page”, the trophi can be classified into nine groups, two of which are not within our study. The types of trophi are divided into malleate trophi, asymmetrical virgate trophi, virgate trophi, incudate trophi, ramate trophi, malleoramate trophi, and forcipate trophi. Among them, malleate trophi and ramate trophi are filter-feeding groups, asymmetrical virgate trophi are a predacious group, virgate trophi are sucking, incudate trophi, and malleoramate trophi, and forcipate trophi are carnivorous [8]. These traits are closely related to their life history and habitat preferences, which can show a clear response of rotifer communities to environmental changes [27,37,38].

### 2.4. Statistical Analysis

The trophic level index (TLIc) is highly correlated with the trophic level and can be used to indicate eutrophication [39]. TLI is a weighted sum based on the correlations between Chl-a, TP, TN, SD, and CODMn [40]. Each formula is established as follows:(1)$$TLI\left(Chl-a\right)=10\left[2.5+1.086\ln(Chl-a)\right]$$
(2)$$TLI\left(TP\right)=10\left[9.436+1.624\ln(TP)\right]$$
(3)$$TLI\left(TN\right)=10\left[5.453+1.694\ln(TN)\right]$$
(4)$$TLI\left(SD\right)=10\left[9.436+1.624\ln(SD)\right]$$
(5)$$TLI\left(CODMn\right)=10\left[5.118+1.94\ln(CODMn)\right]$$

The TLI equation was calculated as follows:(6)$$TLIc=\sum_{j=1}^{m}W_{j}\times TLI\left(j\right)$$
(7)$$W_{j}={rij}^{2}/\sum_{j=1}^{m}{rij}^{2}$$
where Wj represents the correlative weight of the environmental factor, and rij is the correlation coefficient between the reference Chl-a and each parameter, which was obtained from the 26 major lake survey data sets for China [40]. TLIc was calculated using Excel-2019. (TSI < 30 oligotrophic, 30 < TSI < 50, mesotrophic or eutrophication > 50, and eutrophication). Boxplots were used to evaluate differences in TLIc between the different months in the Jinan Section of the Xiaoqing River (Figure 2).

The TLIc between time scales was assessed using SPSS 22, and an analysis of variance (ANOVA) and Tukey tests were performed to determine significant differences (*p* < 0.05). The Bray–Curtis distances for the rotifer community were calculated, as well as distances for each functional group in the Vegan4.2.3 [41] package in Rstudio, which gives equal weight to each trait. The rotifer communities were clustered based on the Bray–Curtis dissimilarity to reflect an aggregated tendency in the Jinan section of the Xiaoqing River and principal coordinate analysis (PCoA) on this distance matrix, which, based on the first two PCoA scores, summarizes the trait data in the functional group.

To test whether eutrophication gradually homogenizes communities, the FD 1.0.12.1 [42] R-package was utilized to compute three common trait diversity measures: (1) functional richness (FRic), which is the number of species in a filled niche space; (2) functional evenness (FEve), which is the extent to which species are distributed in the filled niche space; and (3) functional divergence (FDiv), which is the maximum degree of differentiation of functional characteristics within a community in the niche space [22,43]. All three complement each other and describe the pattern of species distribution within the functional space [44]. To identify responses of the trait diversity of the rotifer community to increased eutrophication, linear regression and Pearson’s correlation analysis were adopted to test the influence of independent variables for TLIc on dependent variables of the functional diversity index implemented in R Studio 4.2.2. using the Vegan 4.2.3. [41] and ggplot 2 3.4.0 packages.

The adespatial 0.3.20 [45] and adegraphics 1.0.21 [46] R packages were used to partition the total beta diversity into its species replacement and richness difference components [28,43]. Species replacement, also known as turnover, refers to the simultaneous gain and loss of species, i.e., functional trait differentiation. Richness difference refers to the differences in the number of functional groups in the different communities, and it may reflect the diversity of niches throughout the study area [47,48,49,50]. Visualization of means of total beta diversity (species replacement and richness difference) and similarity (Sim:1-βtotal) components are shown in triangular plots, where each side of the triangle represents one component. The temporal dissimilarity between all sites was calculated to observe the relationships between all beta diversity and eutrophication between sites. All figures in the text were performed in R Studio4.2.2 using the ggplot2 3.4.0 [51] and corrplot 0.92 [52] packages.

## 3. Results

The annual mean trophic level index (TLIc) in the Jinan section of the Xiaoqing River was determined to be 54.93 ± 6.00 (TSI > 50), signifying a eutrophic condition. The sampling sites were categorized as either mesotrophic or eutrophic, with no oligotrophic sites identified. The measured TLIc in the Jinan section of the Xiaoqing River exhibited significant variation across the 12 months, demonstrating pronounced seasonal fluctuations. TLIc values reached their peak in spring (60.54 ± 4.61) and reached their lowest point in winter (51.16 ± 4.44) (Figure 2).

A total of 46 species of rotifers were identified at all sampling sites during the study period. The main dominant species are *Brachionus diversicornis*, *Brachionus calyciflorus*, *Brachionus angularis*, *Keratella cochlearis*, and *Polyarthra trigla*. The weaker clustering of the Jinan section of Xiaoqing River watersheds indicated the similarity of rotifer communities in different months. (Figure 3) This investigation categorized rotifer taxa into distinct functional groups based on four key traits: trophi, size, feeding habits, and swimming behavior. A principal coordinate analysis (PCoA), utilizing Bray–Curtis distances, effectively discriminated between rotifer species belonging to various functional groups. Notably, taxa associated with size and swimming displayed a dispersed state, implying a tendency toward homogeneity within these two groups (Figure 4). The dynamics of the rotifer communities were stratified according to the functional traits of the four specified functional groups during the 12 sampling periods. On a temporal scale, all groups exhibited distinctly dominant functional taxa (Y ≥ 0.02, with a relative abundance exceeding 20% per month), except for the dominant functional group within the size taxon, which exhibited variation across months, particularly in malleate trophi, filter-feeding, and planktonic (Figure 5).

The results of the Pearson correlation analysis demonstrated a significant negative association between functional richness (FRic) and the trophic level index (TLIc), while functional evenness (FEve) and functional divergence (FDiv) exhibited an extremely significant positive correlation with TLIc (Figure 6). FRic exhibited a positive increase with escalating eutrophication, signifying an augmentation in the number of species within the ecosystem corresponding to an increasing TLIc index. Eutrophication negatively impacted FEve (R^2^ = 0.039, *p* = 0.041), indicating that with the elevation of eutrophication levels, rotifer communities manifested a more irregular structure, tending towards similarity in functional traits. Moreover, FDiv (R^2^ = 0.039, *p* = 0.041) also substantially decreased with increasing eutrophication. This observation underscores the robust ecological niche competition and the spatial loss of functional traits among species, implying a convergence towards similarity in the functional characteristics of rotifers in aquatic ecosystems as eutrophication levels escalate (Figure 7).

A pronounced elevation in total beta diversity was discerned at all sampled locations, indicating substantial taxonomic heterogeneity within rotifer communities. A total of 54.52% was contributed by richness difference (richness difference = 0.398), which was the most important component of beta diversity, while 45.48% was contributed by species replacement (species replacement = 0.332) (Figure 8). Functional beta diversity was used to analyze functional traits of the four functional groups, and the results show that beta diversity was low in all functional groups. This finding implies a heightened uniformity in functional traits among rotifer communities. When the division of functional beta diversity was assessed, a different pattern in the contribution of beta diversity components was observed in functional groups. The richness difference for the trophi functional group (richness difference = 0.236), the feeding functional group (richness difference = 0.0187), and the swimming functional group (richness difference = 0.196) were important components of beta diversity for all rotifers except for the size functional group, where species turnover was not a major factor. The contribution rates of the richness difference were 73.75%, 84.62%, and 91.16%, while species replacement only contributed 26.25%, 15.38%, and 8.84% (Figure 9). The observed heightened similarity among each functional group was ascribed to variations in species abundance. Rotifer communities respond to changing trophic conditions via fluctuations in the relative abundance of different functional traits in functional groups.

## 4. Discussion

The temporal patterns of rotifer community functional groups in a representative urban river were examined using measures of total beta diversity and functional diversity based on rotifer functional groups. This analysis aimed to investigate the impact of eutrophication on the compositional dynamics of rotifer community functional groups. The results revealed a significant influence of eutrophication levels in aquatic ecosystems on the rotifer community’s functional diversity index (FEve and FDiv). Notably, each functional group exhibited relatively low functional beta diversity. Furthermore, the partitioning of beta diversity highlighted that these compositional shifts were predominantly driven by species’ nestedness (richness difference). This homogenization manifests as functional homogenization, driven by the depletion of rare species and the proliferation of more widespread species.

Accelerated urbanization results in a high degree of environmental heterogeneity [53]. Land use shifts have exacerbated eutrophication in production processes, and nitrogen, phosphorus, and other chemicals produced in daily human life and economic production enter aquatic ecosystems through soil erosion and surface runoff, thereby exerting a discernible impact on aquatic organisms [7]. Rotifer species have specific preferences for particular environments, and the tolerances of environmental conditions are manifested in their functional traits [10,54]. The eutrophication of water bodies leads to a proliferation of phytoplankton, which leads to the increase in herbivorous rotifers and their absolute dominance in the ecosystem. Furthermore, species endowed with nutrient preferences conducive to phytoplankton feeding, such as *Polyarthra*, demonstrate a notable advantage, as evidenced by their substantial numerical increase in tandem with the progression of eutrophication [18,55]. Secondly, the nutrient status in the aquatic ecosystem selectively filters out species that are not well adapted to high-nutrient environments. The generation of harmful cyanobacteria often accompanies this process due to eutrophication. Tolerant species, capable of thriving in such conditions, dominate significant ecological niches. These dynamics pose challenges, leading to the homogenization of the rotifer community. Notably, factors like primary productivity can exert an influence on these traits [3,26]. Rotifer communities respond to changing environmental conditions through fluctuations in the relative abundance of different functional groups. However, their complex and specific morphological structure is usually resistant to external influences over time due to rivers’ specific mixing and scouring dynamics and hydrological conditions [8].

Environmental heterogeneity is one of the important factors contributing to beta diversity in freshwater ecosystems [56]. Human disturbances can create living conditions for species with different functional traits to survive. However, eutrophication can exert strong selective pressure on species’ functional traits, leading communities towards functional homogenization [57]. In this study, beta diversity at the functional group level was remarkably low in rotifer communities under eutrophic conditions, contrasting with higher diversity at the taxonomic level. This finding, consistent with prior research, affirms that the variability in environmental conditions within aquatic ecosystems exerts a more pronounced effect on functional groups than individual species [26,58]. Species belonging to the same functional group utilize the same resources in a similar manner [54]. The fitting result obtained from the TLIC and functional divergence (FDiv) at the α-level showed that ecological niche competition becomes more intense with increasing trophic levels.

Consequently, the low values of functional beta diversity may be attributed to functional convergence resulting from competition for similar ecological niches. Only a select few functional groups adapted to the eutrophic environment allowed in the habitat, thereby contributing to the homogenization of the rotifer community [30,59]. Rotifer communities constitute an important link within an aquatic ecosystem’s food chain, and thus, its homogenization may disrupt important ecological functions and services in urban river ecosystems [60,61].

In the Jinan section of the Xiaoqing River rotifer community, the lower functional beta diversity indicates that the community is a functional redundancy. Conversely, the higher species beta diversity indicates that community homogeneity is achieved through functional traits rather than specific taxonomic classes [62]. In the rotifer community of the Jinan section of the Xiaoqing River, the diminished functional beta diversity signifies functional redundancy within the community. In contrast, heightened species beta diversity suggests that community homogeneity is achieved through functional traits rather than specific taxonomic classes [62]. This observation underscores the necessity of considering not only species traits but also functional traits of species when implementing conservation strategies for relevant ecosystems [26].

The higher contribution of richness difference in functional β-diversity is attributed to the absence of rare traits at the functional level and the increase in functionally redundant species. Widespread species replace rare species, and the loss of species with specific traits may decrease beta diversity. Consequently, a greater focus on conserving rare species is favorable for ecosystem restoration [26,63]. From the perspective of rotifer functional communities, which functionally similar species are responsible for the community’s tendency to homogenize functional traits, or which species result from specific functional traits, remains uncertain. This hypothesis necessitates further investigation for comprehensive understanding.

## 5. Conclusions

The findings of this study illustrate that heightened rotifer communities compete more strongly for ecological niches in urban rivers, driven by nutrient levels in the urban river (i.e., declines in functional evenness and functional divergence), and that functional trait loss and loss of functional properties (i.e., RichDiff) are an important component of functional beta diversity. Hence, we suggest that the eutrophication of water bodies in urban rivers leads to a homogenization of rotifer communities in which functional homogeneity assumes a dominant role. Ecosystems that rely on a few species for critical impacts are fragile and more vulnerable to environmental change. Preserving biological heterogeneity is, therefore, critical to the ecological integrity of impaired rivers. Future research endeavors should prioritize an in-depth exploration of additional factors contributing to aquatic community homogenization, employing a comprehensive approach to mitigate further biotic homogenization. 

## Figures and Tables

**Figure 1 biology-12-01488-f001:**
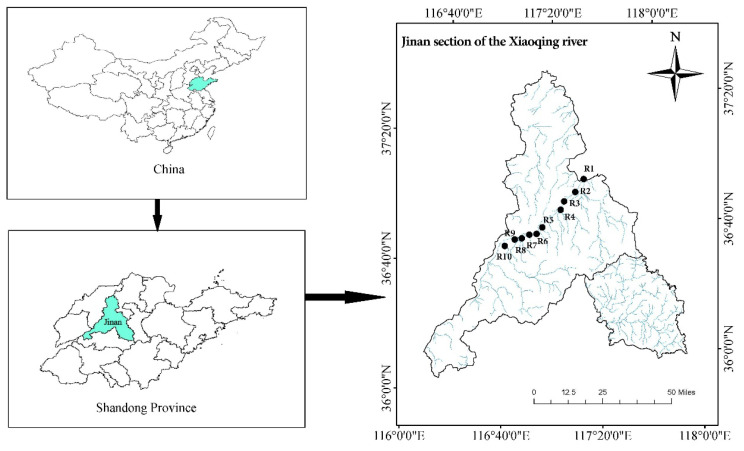
Location of the study area and the sampling sites in the city of Jinan, with the sampling sites marked by black dots.

**Figure 2 biology-12-01488-f002:**
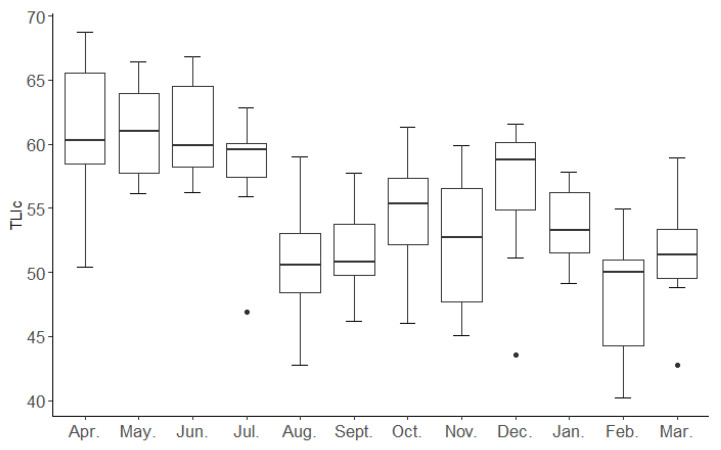
The annual values of the trophic level index (TLIc) in the Jinan section of the Xiaoqing River.

**Figure 3 biology-12-01488-f003:**
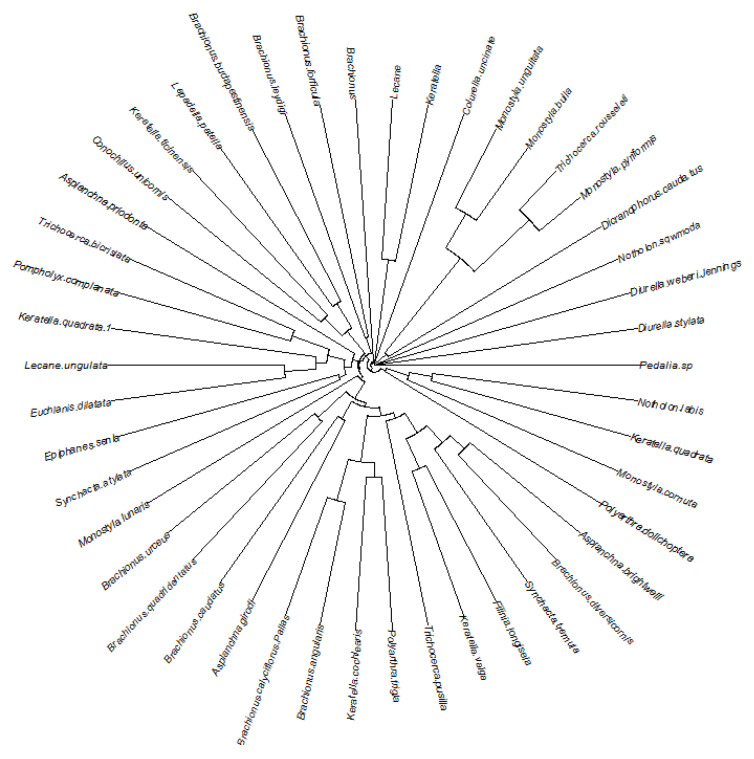
Bray-Curtis dissimilarity in the rotifer communities in the Jinan section of the Xiaoqing River watershed. Basin clustering based on community similarity.

**Figure 4 biology-12-01488-f004:**
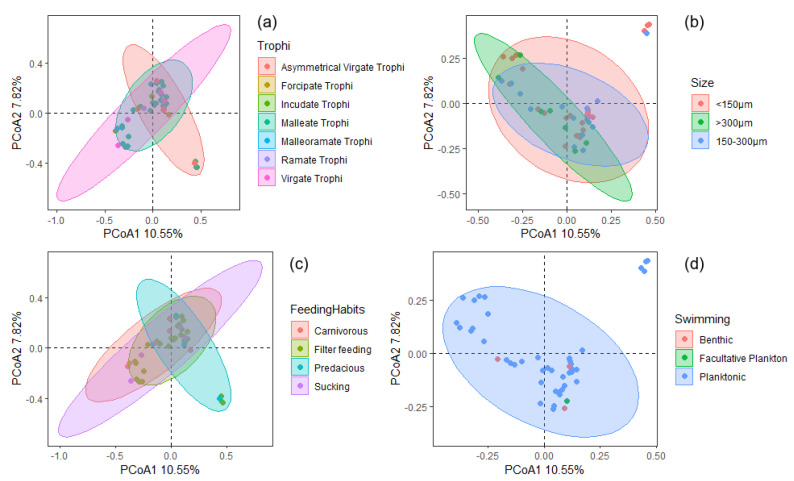
Principal coordinate analysis (PCoA) of 46 rotifer species based on Bray-Curtis distance, (**a**) trophi, (**b**) size, (**c**) feeding habits, and (**d**) swimming.

**Figure 5 biology-12-01488-f005:**
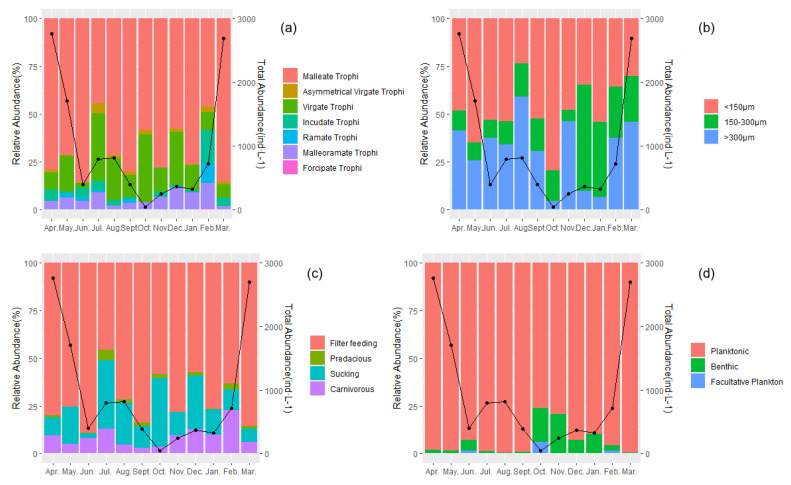
The proportions (*Y*-axis on the left) and the total rotifer abundances (*Y*-axis on the right) in 118 samples collected from the Jinan section of the Xiaoqing River in the twelve months. The lines represent the total rotifer abundances, and the columns represent the rotifer functional trait composition. (**a**) trophi, (**b**) size, (**c**) feeding habits, and (**d**) swimming.

**Figure 6 biology-12-01488-f006:**
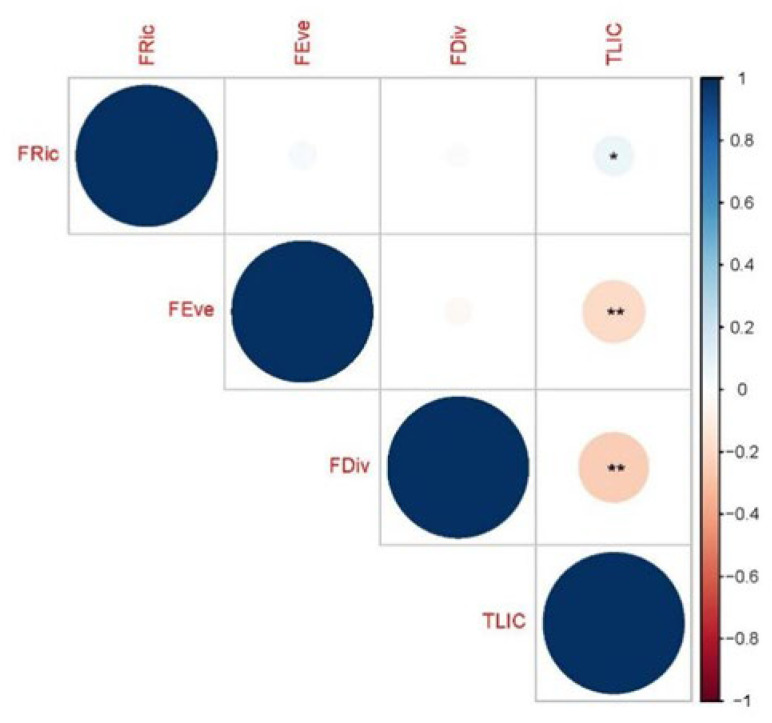
A Pearson correlation analysis of community indicators and a trophic level index (TLIc) for 118 sampling sites in the Jinan section of the Xiaoqing River. * stands for significant correlation; ** stands for extremely significant correlation.

**Figure 7 biology-12-01488-f007:**
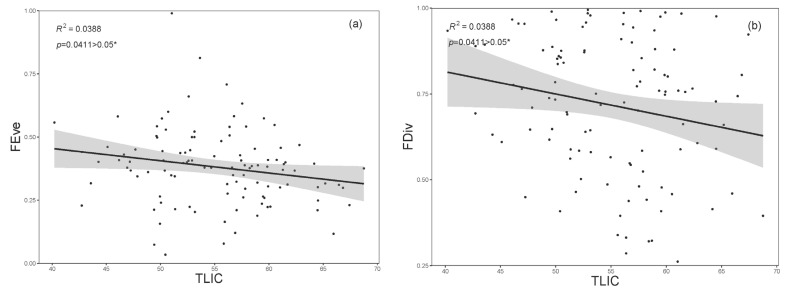
Regression of community indicators and the trophic level index (TLIc) for 118 sampling sites in the Jinan section of the Xiaoqing River, with shaded areas representing 95% confidence intervals. (**a**) Functional evenness and (**b**) functional dispersion, * stands for significant correlation.

**Figure 8 biology-12-01488-f008:**
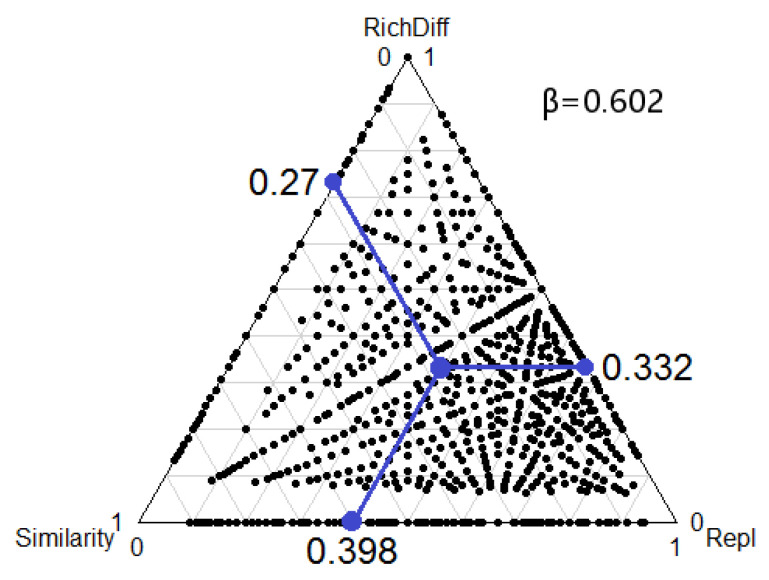
Triangulation of the annual variation relationship of rotifer taxonomic levels from 10 sites in the Jinan section of the Xiaoqing River. Each point (black dot) represents a pair of sites. Its position is determined by three values of similarity (1–D), Repl (replacement), and RichDiff (difference in richness) matrices; each triad sums to 1. The mean of similarity (1–D), Repl (replacement), and RichDiff (richness difference) matrices are represented by the larger point.

**Figure 9 biology-12-01488-f009:**
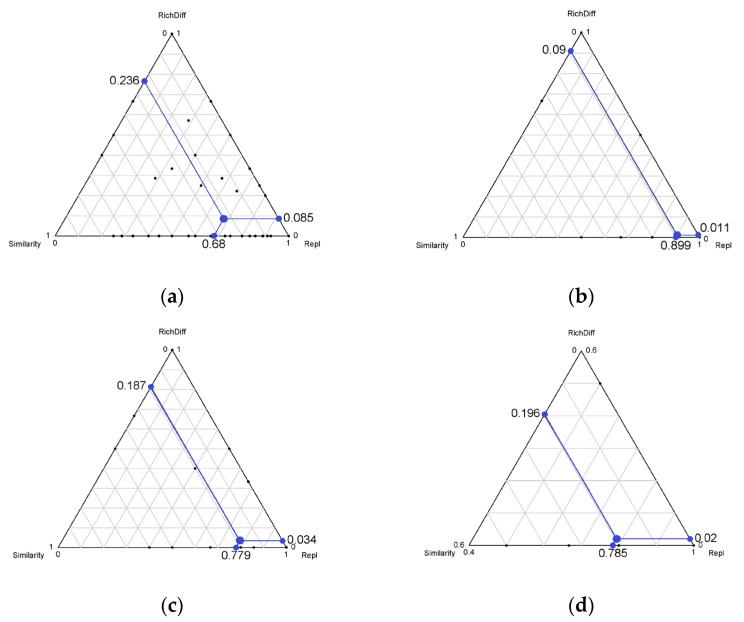
Triangulation of the annual variation relationship of rotifer functional levels from 10 sites in the Jinan section of the Xiaoqing River. (**a**) Trophi, (**b**) size, (**c**) feeding habits, and (**d**) swimming.

**Table 1 biology-12-01488-t001:** Creating functional classifications from species traits that were split into functional groups based on trophi, size, feeding habits, and swimming.

Trait Category	Trait Modality	Typical Species
trophi	(1) malleate trophieoram	*Brachionus calyciflorus*; *Keratella valga*
	(2) asymmetrical virgate trophi	*Diurella weberi*; *Trichocerca pusilla*
	(3) virgate trophi	*Polyarthra trigla*; *Synchaeta tremula*
	(4) incudate trophi	*Asplanchna girodi*
	(5) ramate trophi	*Conochilus unicornis*
	(6) malleoramate trophi	*Pompholyx complanata*
	(7) forcipate trophi	*Dicranophorus caudatus*
size	(1) <150 μm	*Keratella valga; Brachionus urceus*
	(2) 150–300 μm	*Brachionus leydigi; Monostyla lunaris*
	(3) >300 μm	*Asplanchna priodonta*; *Epiphanes senta*
feeding habits	(1) filter-feeding	*Brachionus calyciflorus*; *Conochilus unicornis*
	(2) predacious	*Asplanchna brightwelli*; *Asplanchna priodonta*
	(3) sucking	*Synchaeta atylata; Polyarthra trigla*
	(4) carnivorous	*Trichocerca pusilla; Diurella weberi*
swimming	(1) planktonic	*Brachionus angularis; Monostyla bulla*
	(2) benthic	*Brachionus urceus; Euchlanis dilatata*
	(3) Facultative plankton	*Epiphanes senta*

## Data Availability

Publicly available datasets were analyzed in this study. This data can be found here: (https://pan.baidu.com/s/15kbKKGhZWHVjYmj_k0iU3Q, extraction code: 5ina).

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
