# Peer review of "Homogenization of Functional Diversity of Rotifer Communities in Relation to Eutrophication in an Urban River of North China"

_biology, 2023, doi:10.3390/biology12121488_

Round 1

Reviewer 1 Report

Comments and Suggestions for Authors

1.      The manuscript format is PDF, so it cannot be revised with tracked changes.

2.      Two hypotheses were listed in the last paragraph of the introduction. This is good for a hypothesis-driven study. The authors should offer more information or explanation of the second hypothesis ‘the homogenization in a community is the result of widely distributed species complementation’ in the discussion.

3.      Check punctuations and spaces (e.g. Line 118 …and pH, were measured in situ at the sampling sites.)

4.      Pay attention to the capitalization of letters (e.g. Line 151 Where Wj represents the correlative…)

5.      Line 153 The name of analysis software should be added.

Author Response

1.The manuscript format is PDF. so it cannot be revised with tracked changes.

This has been done. The manuscript was uploaded using DOCX to ensure modification tracking

2. Two hypotheses were listed in the last paragraph of the introduction. This is good for a hypothesis-driven study. The authors should offer more information or explanation of the second hypothesis ‘the homogenization in a community is the result of widely distributed species complementation’ in the discussion.

This has been done. Based on the reviewers' comments. we have added additional information to the revised manuscript.

3. Check punctuations and spaces (e.g. Line 118 …and pH, were measured in situ at the sampling sites.)

This has been done. The entire manuscript was checked for punctuation and spacing.

4. Pay attention to the capitalization of letters (e.g. Line 151 Where Wj represents the correlative…)

This has been done. The entire manuscript has been checked for capitalization.

5.  Line 153 The name of analysis software should be added.

This has been done. The manuscript has added the name of the analysis software at lines 153 to 154,186 to 188.

Reviewer 2 Report

Comments and Suggestions for Authors

The manuscript is a novel conceptual and methodological proposal for the calculation and changes in time and space of beta diversity in aquatic ecosystems, and how eutrophication affects it.

I suggest some minor corrections to be made in the style of using spaces between words, for example the words "Figure3".

Another example: taking care of the spaces and punctuation at the end of each sentence and pointing out a figure, or a quote. For example on line 199 it says "...to be homogeneous. (Figure4) The dynamics..." and it should say "...to be homogeneous (Figure 4). The dynamics..."

Those minor editorial issues should be corrected throughout the document. Some lines where I observed it are 194, 199, 205, 208, 217, 240, 250, 260, 265, 278, 282, 283, 292, 314, 318.

Some minor issues should also be fixed in the References section, on lines 384-385; 400-401. On line 416 it seems that something is missing.

Author Response

Question 1: I suggest some minor corrections to be made in the style of using spaces between words, for example, the words "Figure3".

Answer to question 1: This has been done. The entire manuscript has been checked for punctuation and spaces.

Question 2: Another example: taking care of the spaces and punctuation at the end of each sentence and pointing out a figure, or a quote. For example on line 199 it says "...to be homogeneous. (Figure4) The dynamics..." and it should say "...to be homogeneous (Figure 4). The dynamics...

Answer to question 2: This has been done. The entire manuscript has been checked for punctuation and spaces.

Question 3: Those minor editorial issues should be corrected throughout the document. Some lines where I observed it are 194, 199, 205, 208, 217, 240, 250, 260, 265, 278, 282, 283, 292, 314, 318.

Answer to question 3: This has been done. Editorial issues have been corrected throughout the manuscript.

Question 4: Some minor issues should also be fixed in the References section, on lines 384-385; 400-401. On line 416 it seems that something is missing.

Answer to question 4: Checked and corrected incorrect reference formats in the manuscript.

Reviewer 3 Report

Comments and Suggestions for Authors

The study pointed to the current problem that environmental pollution can significantly reduce species diversity. The research is very interesting and raises the important issue of the impact of eutrophication on rotifer biodiversity. According to Authors, widespread species replace rare species, and the loss of species with specific traits  may lead to a decrease in beta diversity.

I have been conducting research on the presence of rotifers in sewage treatment plants and in the outflow for over two years and I noticed that their occurrence is related to seasonality wastewater treatment technology. In my opinion, far-reaching conclusions cannot be drawn based on a period of 12 months. In addition, it does not mention whether treated wastewater is discharged into the river, which is a source of rotifers and can affect the results. In my opinion, it would be necessary to analyze at least another six months to notice the trend of decrease in diversity.

Other comments:

1. In the abstract results and novelty should be highlighted.

2. Materials and methods: “Species identification was carried out on according to Chinese freshwater rotifer and Guides to the Identifi cation of the Microinvertebrates of the Continental waters of the World”. Please, complete the most important information about the research methodology. 1 ml of sample were placed on microscopic slide? How many repetitions were performed?

3. Figures are clear and well prepared but of poor quality (resolution). Fig. 2 – axes, no explanation of what statistical parameters were marked on the figure. Was the TILC determined on the basis of 10 measurements (R1-R10?) for individual months?

4. Editing errors: Lines: 121,95,137-144, 400-401,416

Author Response

Question 1: I have been conducting research on the presence of rotifers in sewage treatment plants and in the outflow for over two years and I noticed that their occurrence is related to seasonality wastewater treatment technology. In my opinion, far-reaching conclusions cannot be drawn based on a period of 12 months. In addition, it does not mention whether treated wastewater is discharged into the river, which is a source of rotifers and can affect the results. In my opinion, it would be necessary to analyze at least another six months to notice the trend of decrease in diversity.

Answer to question 1: The inspection and rectification cases of the Ministry of Ecology and Environment of the People's Republic of China show that since 2017, the Xiaoqing River Basin has implemented comprehensive prevention and control management at the source. 66 industrial enterprises were relocated, 7,190 enterprises that illegally discharged pollutants were banned, and 8 water-related enterprises were rectified. The daily discharge of industrial polluted wastewater increased from 60,000 tons/day. Reduced to 0.5 tons/day, industrial pollution is small. Therefore, this manuscript did not consider the impact of industrial wastewater discharge on rotifer community structure.

Question 2: In the abstract results and novelty should be highlighted.

Answer to question 2: This has been done. This section has been added to the abstract of the manuscript. Please see lines 21 to 25, in the revised manuscript for details.

Question 3: Materials and methods: “Species identification was carried out on according to Chinese freshwater rotifer and Guides to the identification of the Microinvertebrates of the Continental waters of the World”. Please, complete the most important information about the research methodology. 1 ml of sample were placed on microscopic slide? How many repetitions were performed?

Answer to question 3: This has been done. 1 ml of sample was placed on a microscopic slide for full slide observation and twice repetitions were performed. Please see lines 112 to 114in the revised manuscript for details.

Question 4: Figures are clear and well prepared but of poor quality (resolution). Fig. 2 – axes, no explanation of what statistical parameters were marked on the figure.

Answer to question 4: This has been done. All Figures in the manuscript have been replaced, and the statistical parameters in Figure 2 have been added.

Question 5: Was the TILC determined on the basis of 10 measurements (R1-R10?) for individual months?

Answer to question 5: TLIc is calculated based on 10 sampling points in each month, of which the R6 point in June and the R8 point in December were not sampled.

Question 6: Editing errors: Lines: 121,95,137-144, 400-401,416

Answer to question 6: This has been done. Editorial issues have been corrected throughout the manuscript.

Reviewer 4 Report

Comments and Suggestions for Authors

I am very sceptical about what can be learnt from inspecting each trait individually. Species possess a suite of traits that interact and should be analysed as such (i.e. all traits together)

Abstract

L14: please write “beta diversity” throughout the manuscript and not Beta diversity

L17: paste tense

Introduction

L31-35: split this long sentence into two; to what is “their” on L 33 referring to?

L38: delete “the”

L42: the sentence starts with Although, but then the secondary sentence is missing -> please rephrase

L44: you start with “but” without providing a primary sentence -> rephrase; I guess that the sentence on L42 is part of the sentence on L44; if yes, please connect both and pay attention not to write too long sentences

L49: “Eutrophication reduces habitat as a cause of reduced biodiversity, “ -> rephrase because impossible to understand

L53: space between only and at

L65-66: rewrite the sentence because its meaning is unclear

L66-69: please write species richness and not RDiff and write replacement and nor Repl because the abbreviations stem from the R package and are method related and are not appropriate for an introduction

L70: please rephrase because unclear

L71-75: it is unclear if you are referring to taxonomic or functional beta diversity, and therefore I suggest that you write taxonomic or functional whenever it is written about beta diversity

L85: please write “and is the result …”

L85: please explain what do you mean with species complementation?

Material & Methods

L102-103: unclear; you took 50 L and did not filter it? You mixed the sample underwater? Rom which depth did you take the sample?

L108-109: please provide a reference for both books

L120: what is a trophi tree? Please correct

Table1: typos for Epiphanes senta; Synchaeta stylata

L121-124: split the sentence into two

L124-127: this is definitely the wrong citation for this sentence; please, furthermore consider and discuss Gilbert, J.J., 2022. Food niches of planktonic rotifers: Diversification and implications. Limnology and Oceanography67(10), pp.2218-2251.

L135: please provide an example about which number indicate good or bad ecological status (e.g. values > 50 = eutroph)

L152: what are you meaning with “time scales”? please, clarify

L152: please rephrase that you applied ANOVA using SPSS

L154: which distance did you ues for functional beta diversity?; rephrase the sentence because “which gives equal wight to it” is not the “fault” of vegan but it depends on your choice as researchers

L158: delete “which generated point plot” because PCoA is sufficiently known among ecologists

L155/160/171/170: whenever you use a R package, you have to cite the developers (e.g. vegan - Oksanen et al. )

L161: please write “the amount of niche space filled by species in the community (functional richness)

L165: please write “these indices “ instead of “they all three”

L167-170: it is unclear what is the dependent and the independent variable in linear regression and correlation analysis -> rephrase (please use short sentences that are easy to follow)

Results

L189: please do ANOVA to see if these differences are statistically significant

L191: “were” instead of “are”

L194: delete “from the Bray-Curtis distance”

Figure 3: typos in species names; please clarify what this figure is: is this a cluster analysis of similarity of species based on their traits?

L194-197: this is method description and should eb integrated in the method section

Figure 4: how is it possible that 4 different PCoAs have the same variability explained for the two axes? Please explain if this is just one analysis but you colour the points according to 4 different characteristics; if this is true, I am still wondering why a) and c) are similar and b) and d) are quite similar regarding point location but not all 4 are similar

Figure 4: it is completely unclear what the input variables are for this plot

Figure 6: please write * p> 0.05, ** p> 0.01, *** P< 0.001

Figure 7: considering that (a) is not statistically significant, there is no need to show nay relationship (please delete the line in the plot)

L214-217: I do not agree with you that a R2 of 3% is something meaningful; furthermore, did you control residuals for any heterogeneity? In addition, it seems that values are bounded between 0 and 1 and therefore gaussian regression is not appropriate; please try with beta regression

L237: sampling sites

Figure 8: taxonomic beta diversity was quite variable, and therefore also state the range of values; the triangulation is about the mean value but you should also show the median with such variable data

L237-250: please write about functional and taxonomic beta diversity

L253-256: this is discussion and should be moved to the discussion section

L252: which analysis showed that?

Figure 9: why are there less points in figure 9 than in figure 8? Both figures are based on the same number of comparisons of samples and should therefore show the same number of points

Discussion

L270-272: I do not agree that you have the results to state this; your results are too weak to state this

Considering that I have serious concerns about your data analyses and results, I cannot really evaluate the discussion section

Comments on the Quality of English Language

has to be improved

Author Response

Question 1: I am very skeptical about what can be learned from inspecting each trait individually. Species possess a suite of traits that interact and should be analyzed as such (i.e. all traits together) ‘’

Answer to question 1: Analysis based on individual characteristics is considered scientific and reasonable and can be used for biological diversity. For the analysis of individual traits, reference was made to this literature. Marcacci G, Westphal C, Wenzel A, Raj V, Nolke N, Tscharntke T, et al. Taxonomic and functional homogenization of farmland birds along an urbanization gradient in a tropical megacity. Glob Chang Biol. 2021;27(20):4980-4994.

Question 2: Please write “beta diversity” throughout the manuscript and not Beta diversity.

Answer to question 2: This has been done.

Question 3: L17paste tense

Answer to question 3: This has been done.

Question 3: L31-35: split this long sentence into two; to what is “their” on L 33 referring to?

Answer to question 3: This has been done, please see lines 39 to 42, in the revised manuscript for details. “their” on L 33 referring to the chemical composition of aquatic ecosystems.

Question 4: L38: delete “the”

Answer to question 4: This has been done. Please see line 45, in the revised manuscript for details.

Question 5: L42: the sentence starts with Although, but then the secondary sentence is missing -> please rephrase

Answer to question 5: This has been done. Please see lines 52 to 53 in the revised manuscript for details.

Question 6: L44: you start with “but” without providing a primary sentence -> rephrase; I guess that the sentence on L42 is part of the sentence on L44; if yes, please connect both and pay attention not to write too long sentences

Answer to question 6: This has been done. Please see lines 52 to 53 in the revised manuscript for details.

Question 7: L49: “Eutrophication reduces habitat as a cause of reduced biodiversity, “ -> rephrase because impossible to understand

Answer to question 7: This has been done. Please see lines 57 to 58 in the revised manuscript for details.

Question 8: L53: space between only and at

Answer to question 8: This has been done.

Question 9: rewrite the sentence because its meaning is unclear

Answer to question 9: This has been done.

Question 10: L66-69: please write species richness and not RDiff and write replacement and nor Repl because the abbreviations stem from the R package and are method-related and are not appropriate for an introduction

Answer to question 10: This has been done.

Question 11: L70: please rephrase because unclear

Answer to question 11: This has been done.

Question 12: L71-75: it is unclear if you are referring to taxonomic or functional beta diversity, and therefore I suggest that you write taxonomic or functional whenever it is written about beta diversity

Answer to question 12: This has been done.

Question 13: L85: please write “and is the result …”

Answer to question 13: This has been done.

Question 14: L85: please explain what do you mean with species complementation?

Answer to question 14: Species complementation is the process by which common species supplement rare species.

Question 15: L102-103: unclear; you took 50 L and did not filter it? You mixed the sample underwater? Rom which depth did you take the sample?

Answer to question 15: On each sampling date, 50 L water was collected at 1.5 meters underwater at each sampling site using a 5 L Specification sampler. The supernatant was concentrated to 30 ml after settling in a cylindrical separatory funnel for 24 h to quantify rotifers.

Question 16: L108-109: please provide a reference for both books

Answer to question 16: Wang Jiaji. Records of freshwater rotifers in China. Beijing: Science Press, Dumont, H. J. F . Guides to the Identification of the Microinvertebrates of the Continental Waters of the World.

Question 17: L120: what is a trophi tree? Please correct

Answer to question 17: This has been done. Misspelled, trophi is the chewer of rotifers

Question 18: Table1: typos for Epiphanes senta; Synchaeta stylata

Answer to question 18: This has been done.

Question 19: L121-124: split the sentence into two

Answer to question 19: This has been done.

Question 19: L124-127: this is definitely the wrong citation for this sentence; please, furthermore consider and discuss Gilbert, J.J., 2022. Food niches of planktonic rotifers: Diversification and implications. Limnology and Oceanography, 67(10), pp.2218-2251.

Answer to question 19: This has been done.

Question 20: L135: please provide an example about which number indicate good or bad ecological status (e.g. values > 50 = eutroph)

Answer to question 20: This has been done.

Question 21: what are you meaning with “time scales”? please, clarify

Answer to question 21: The time scale refers to the changes in community structure over time.

Question 22: L152: please rephrase that you applied ANOVA using SPSS

Answer to question 22: ANOVA analysis was conducted on the data using SPSS.

Question 23: L158: delete “which generated point plot” because PCoA is sufficiently known among ecologists

Answer to question 23: This has been done.

Question 24: L155/160/171/170: whenever you use a R package, you have to cite the developers (e.g. vegan - Oksanen et al. )

Answer to question 24: This has been done.

Question 25: Please write “the amount of niche space filled by species in the community (functional richness)”

Answer to question 25: This has been done.

Question 25: Please write “the amount of niche space filled by species in the community (functional richness)”

Answer to question 25: This has been done.

Question 26: L165: please write “these indices “ instead of “they all three”

Answer to question 26: This has been done.

Question 27: L167-170: it is unclear what is the dependent and the independent variables in linear regression and correlation analysis -> rephrase (please use short sentences that are easy to follow)

Answer to question 27: This has been done. The independent variable and dependent variable are TLIC and functional diversity.

Question 28: L189: please do ANOVA to see if these differences are statistically significant

Answer to question 28: Statistical analysis has been conducted and the difference results are statistically significant.

Question 29: L191: “were” instead of “are” L194: delete “from the Bray-Curtis distance”

Answer to question 29: This has been done.

Question 30: L191: Figure 3: typos in species names; please clarify what this figure is: is this a cluster analysis of the similarity of species based on their traits?

Answer to question 30: This has been done. This a cluster analysis of the similarity of species based on their traits.

Question 31: L194-197: this is the method description and should be integrated in the method section

Answer to question 31: This has been done.

Question 32: Figure 4: how is it possible that 4 different PCoAs have the same variability explained for the two axes? Please explain if this is just one analysis but you colour the points according to 4 different characteristics; if this is true, I am still wondering why a) and c) are similar and b) and d) are quite similar regarding point location but not all 4 are similar Figure 4: it is completely unclear what the input variables are for this plot

Answer to question 32: Figure 4 shows the PCoA analysis of four different functional traits, among which the similarity between (a) and (c) is due to the correlation between the trophi and feeding habits of rotifers, and the similarity between (b) and (d).

Question 33: Figure 6: please write * p> 0.05, ** p> 0.01, *** P< 0.001 , Figure 7: considering that (a) is not statistically significant, there is no need to show any relationship (please delete the line in the plot)

Answer to question 33: This has been done.

Question 34: L214-217: I do not agree with you that an R2 of 3% is something meaningful; furthermore, did you control residuals for any heterogeneity? In addition, it seems that values are bounded between 0 and 1 and therefore gaussian regression is not appropriate; please try with beta regression

Answer to question 34: Although the R-value is small, the P-value is less than 0.05, indicating a highly significant difference. Still trying beta regression, which will be reflected in the next modification.

Question 35: Figure 8: taxonomic beta diversity was quite variable, and therefore also state the range of values; the triangulation is about the mean value but you should also show the median with such variable data

Answer to question 35: The blue dots in the picture represent the median.

Question 36: L253-256: this is a discussion and should be moved to the discussion section

Answer to question 36: This has been done.

Question 37: Figure 9: why are there fewer points in Figure 9 than in Figure 8? Both figures are based on the same number of comparisons of samples and should therefore show the same number of points

Answer to question 37: Figure 8 is analyzed based on the taxonomic level of rotifers, while Figure 9 is analyzed based on the functional level of rotifers. Therefore, Figure 9 has fewer points than Figure 8.

Question 38: L270-272: I do not agree that you have the results to state this; your results are too weak to state this Considering that I have serious concerns about your data analyses and results, I cannot really evaluate the discussion section

Answer to question 38: We have provided an explanation for your question and strengthened the narrative of the discussion. We sincerely hope that you can take the time to provide suggestions for the discussion section. Thank you very much.

Round 2

Reviewer 3 Report

Comments and Suggestions for Authors

The first question on the length of the observation period is not answered, which is crucial to the entire manuscript. The resolution of the drawings is still low, especially for Figure 3.

Author Response

Question 1: The first question on the length of the observation period is not answered, which is crucial to the entire manuscript.

Answer to question 1: Referring to some papers on rotifers, many scientific hypotheses were validated through annual surveys, thus conducting annual surveys of rotifer communities.

Question 2: The resolution of the drawings is still low, especially for Figure 3..

Answer to question 2: The images in the manuscript have been replaced and the original images will be provided in the attachment for review.

Reviewer 4 Report

Comments and Suggestions for Authors

L59-60: This sentence is impossible to understand

L62-64: This sentence is impossible to understand

L75: delete “among beta diversity”

L80: This sentence is impossible to understand

L86: please write “influences” instead of “contributes to” and write “leads to” instead of “tends to”

L131: please write Table and not Tabel

L137: wrong ciation for this sentence; 8 is about sediments, use 26

L160-167: correct hanging letters (this is not how words are spilt); but now I see that his happens all the text down -> correct

L180: vegan needs a citation

L181: ggforce needs a citation

L206: . (full stop) is missing

Table 1/figure 3 : please write Synchaeta tremula; Synchaeta stylata

Figure 3 seems to be used for clustering of reservoirs but there is no labelling of the Jinan versus Xiaoqing; please make a simple vertical graph with clear indication of the sections/groups

L211: not taxa but groups

Figure 4 is strange: why are the two left and the two right plots, respectively the same regarding point position? But the left and right plots are different even though the var explained (%) values are the same? Are you performing a PCoA with the traits of all species? If yes, then the points of all 4 plots should have the same x-y coordinates; please, clarify

L214: what are you trying to describe here? Malleate, filter-feeding and planktonic species are dominating while for size it is less clear

Figure 6: I do not agree with you that a R2 of 3% is something meaningful; furthermore, did you control residuals for any heterogeneity? In addition, it seems that values are bounded between 0 and 1 and therefore gaussian regression is not appropriate; please try with beta regression. These results are completely borderline and when I consider these p-values as significant, their R- values are so low that this only is a random signal.

For figure 9 and 8 you write that you have 10 sites and you show comparisons between sites: with pairwise comparisons, you have k*(k-1)/2 comparisons = 45 points in your case; where are all those points in figure 9?

Figure 9: why are there less points in figure 9 than in figure 8? Both figures are based on the same number of comparisons of samples and should therefore show the same number of points

Comments on the Quality of English Language

several sentences are strange and I only corrected the worst sentences

Author Response

Question 1:L59-60: This sentence is impossible to understand

Answer to question 1: This has been done.

Question 2: L62-64: This sentence is impossible to understand

Answer to question 2: This has been done.

Question 3:  L75: delete “among beta diversity”

Answer to question 3: This has been done.

Question 4: L80: This sentence is impossible to understand

Answer to question 4: This has been done.

Question 5: L86: please write “influences” instead of “contributes to” and write “leads to” instead of “tends to”

Answer to question 5: This has been done.

Question 6: L131: please write Table and not Tabel

Answer to question 6: This has been done.

Question 7: L137: wrong ciation for this sentence; 8 is about sediments, use 26

Answer to question 7: I didn't understand the meaning of this review.

Question 8: L160-167: correct hanging letters (this is not how words are spilt); but now I see that his happens all the text down -> correct

Answer to question 8: I didn't understand the meaning of this review.

Question 8: L180: vegan needs a citationL181: ggforce needs a citation

Answer to question 8: This has been done.

Question 9: L206: . (full stop) is missing

Answer to question 9: This has been done.

Question 9: L206: . Table 1/figure 3 : please write Synchaeta tremula; Synchaeta stylata

Answer to question 9: This has been done.

Question 10: L206: . Table 1/figure 3 : please write Synchaeta tremula; Synchaeta stylata

Answer to question 10: Figure 3 shows the cluster analysis of all species in the entire section of the Xiaoqing River in Jinan.

Question 11: L211: not taxa but groups

Answer to question 11: This has been done.

Question 12: Figure 4 is strange: why are the two left and the two right plots, respectively the same regarding point position? But the left and right plots are different even though the var explained (%) values are the same? Are you performing a PCoA with the traits of all species? If yes, then the points of all 4 plots should have the same x-y coordinates; please, clarify

Answer to question 12: fig.4 is performing a PCoA with the  different traits of all species. Classified according to different functional traits, different species types and densities in the functional group result in different horizontal and vertical coordinates of points.

Question 13: L214: what are you trying to describe here? Malleate, filter-feeding and planktonic species are dominating while for size it is less clear

Answer to question 13: The meaning I want to express is that only the classification method of rotifer body size does not have an absolutely dominant functional groups, while other classification methods have absolutely dominant functional groups.

Question 14: Figure 6: I do not agree with you that a R2 of 3% is something meaningful; furthermore, did you control residuals for any heterogeneity? In addition, it seems that values are bounded between 0 and 1 and therefore gaussian regression is not appropriate; please try with beta regression. These results are completely borderline and when I consider these p-values as significant, their R- values are so low that this only is a random signal.

Answer to question 14: Beta regression was performed on the data, where the P-values of FEve and TLIc were 7.82e-15 and the R2 were 0.02456,and the P-values of FDiv and TLIc were 4.58e-15 and the R2 were 0.02249.

Question 14: Figure 8 is analyzed based on the taxonomic level of rotifers, while Figure 9 is analyzed based on the functional level of rotifers. Therefore, Figure 9 has fewer points than Figure 8.

Answer to question 14: Figures 8 and 9 are pairwise comparisons based on species level.Figure 8 is analyzed based on the taxonomic level of rotifers, while Figure 9 is analyzed based on the functional level of rotifers. Therefore, Figure 9 has fewer points than Figure 8.